# Unveiling Oxidative Stress-Induced Genotoxicity and Its Alleviation through Selenium and Vitamin E Therapy in Naturally Infected Cattle with Lumpy Skin Disease

**DOI:** 10.3390/vetsci10110643

**Published:** 2023-11-07

**Authors:** Waqas Ahmad, Adeel Sattar, Mehmood Ahmad, Muhammad Waqar Aziz, Asif Iqbal, Muhammad Yasin Tipu, Rana Muhammad Zahid Mushtaq, Naeem Rasool, Hafiz Saleet Ahmed, Muhammad Ahmad

**Affiliations:** 1Department of Pathology, University of Veterinary and Animal Sciences, Lahore 54000, Pakistan; 2Livestock and Dairy Development Department Punjab, Lahore 54000, Pakistan; 3Department of Pharmacology and Toxicology, University of Veterinary and Animal Sciences, Lahore 54000, Pakistan; 4Department of Pharmacology and Toxicology, Faculty of Veterinary and Animal Sciences, The Islamia University of Bahawalpur, Bahawalpur 63100, Pakistan; 5Institute of Microbiology, University of Veterinary and Animal Sciences, Lahore 54000, Pakistan; 6Department of Parasitology, Riphah International University, Lahore 54000, Pakistan; 7Division of Infection Medicine, College of Medicine, The University of Edinburgh, Edinburgh EH16 4SB, UK; 8Department of Livestock Management, Faculty of Veterinary and Animal Sciences, The Islamia University of Bahawalpur, Bahawalpur 63100, Pakistan

**Keywords:** Lumpy skin disease, Pakistan, comet assay, tocopherol, selenium, oxidative stress, genotoxicity

## Abstract

**Simple Summary:**

The present study investigated Lumpy Skin Disease (LSD) outbreaks in cattle, examining the virus’s genotoxic impact and antioxidant therapy efficacy. PCR-confirmed infected cattle were divided into control and treated groups. Selenium and vitamin E were administered to the treated group. Post-therapy, there were notable improvements in hematological indices and reduced serum nitric oxide and malondialdehyde levels. A strong correlation was found between serum nitric oxide, malondialdehyde, and genetic damage. Ordinal logistic regression analysis indicated that substantially improved genotoxicity in treated animals. In conclusion, LSD induces oxidative stress and genotoxicity in cattle, effectively mitigated by Selenium and vitamin E therapy.

**Abstract:**

Lumpy skin disease (LSD) is a contagious infection of cattle caused by a virus of the Poxviridae family, genus Capripoxvirus. In Pakistan, recent outbreaks have resulted in significant nationwide mortality and economic losses. A 20-day prospective cohort study was performed on sixty infected cattle with the aim to evaluate LSD-induced oxidative stress’s genotoxic role and to determine the ameliorative effect of antioxidant therapy using principal component analysis (PCA) and a multivariable ordinal logistic regression model. LSDV was identified from scab samples and nodular lesions using RPO30-specific gene primers. The infected cattle were divided into control and treated groups. The animals were observed initially and finally on day 20 to evaluate the homeostatic, oxidative, and genotoxic changes. The animals in the treated group were administered a combination of selenium (Se) and vitamin E at the standard dose rate for five consecutive days. A substantial (*p* < 0.05) improvement in the hematological indices was observed in the treated group. The treated group also showed a significant (*p* < 0.05) reduction in levels of serum nitric oxide (NO) and malondialdehyde (MDA) post-therapy. The PCA at the final sampling data of the treated group showed that Principal Component (PC1 eigenvalue 1.429) was influenced by superoxide dismutase (SOD; 0.3632), catalase (CAT; 0.2906), and glutathione (GSH; 0.0816) and PC2 (eigenvalue 1.200) was influenced by CAT (0.4362), MDA (0.2056), and NO (0.0693). A significant correlation between serum NO (76%) and MDA levels (80%) was observed with genetic damage index (GDI) scores. The ordinal logistic regression model regarding the use of antioxidant therapy revealed 73.95-times (95%CI; 17.36–314.96) improvement in the GDI in treated animals. The multivariable ordinal logistic regression showed that each unit increase in NO and MDA resulted in a 13% increase in genotoxicity in infected individuals. In conclusion, our study revealed that LSD-induced oxidative stress and lipid peroxidation product causes genotoxicity in affected animals. Furthermore, the combined Se and vitamin E therapy significantly alleviated oxidative stress and genotoxicity in LSD-affected cattle.

## 1. Introduction

LSD is a highly contagious infection of the Capri pox genus that primarily affects cattle and buffaloes. This transboundary virus has recently reemerged as a major issue [1]. The livestock industry suffers significant economic losses [2]. The World Organization for Animal Health has classified LSD as a disease with substantial impact [3]. 

In Pakistan, a total of 190,000 cases of LSD have been documented across the country, resulting in a significant case fatality with more than 7500 deaths. Conversely, a substantial number of 141,000 animals have recovered from the infection [4]. Potential financial ramifications for Pakistan due to LSD can be extrapolated from Ethiopian statistical data. The median economic deficit attributable to each deceased animal in Ethiopia was evaluated to be USD 375. Additionally, there was an estimated USD 141 loss in milk production for every cow affected by the disease. By drawing parallels from these figures, one might infer the prospective economic consequences for Pakistan in the face of an LSD outbreak. [5].

LSD mainly spreads through blood-sucking arthropods such as stable flies and ticks [2]. Direct contact between healthy and infected animals is not considered the transmission route of infection [6]. This condition is characterized by fever, extreme lethargy, and skin nodules that spread to the mucous membranes and internal organs [7,8,9]. Lesions in internal organs result in gastrointestinal and respiratory issues such as ruminal atony, dyspnea, and pneumonia [10]. It is common for outbreaks to occur in epidemics [11]. However, unexplainable gaps of several years between epidemics, irrespective of factors, are commonly observed [12]. The infection alters hematological and biochemical parameters while triggering oxidative stress [13,14]. Oxidative damage is a leading cause of many acute and chronic disorders resulting from the overproduction of free radicals and weakened antioxidant capacity [15]. Elevated levels of oxidative stress and lipid peroxidation biomarkers have been observed in the infected animals [13,14,16].

Furthermore, the oxidative stress itself and through induced lipid peroxidation products, including MDA, further results in genotoxicity in the body [17,18]. 

Among the various antioxidants, vitamin E and selenium (Se) exhibit a unique feature in that they function synergistically to combat lipid peroxides (ROOH). Vitamin E can directly engage with lipid free radicals (ROO•), while Se is an essential trace mineral [19] which serves as the principal co-factor for glutathione peroxidase (GPx) which is an essential enzyme facilitating the conversion of lipid peroxides into inert compounds [20]. Using these natural antioxidants has been shown to mitigate the negative consequences of the overproduction of reactive oxygen species (ROS) [21]. Both Se and vitamin E have been specifically aimed at enhancing the metabolism, reducing oxidative stress, and bolstering the immune and anti-inflammatory responses in dairy cattle [22]. Vitamin E has been also been reported to have the capability of reducing genotoxic effects [23].

However, to our knowledge, limited information is available regarding the impact of oxidative stress levels and lipid peroxidation products on the genotoxicity of blood lymphocytes of LSDV-infected animals. Therefore, the present study was designed to evaluate oxidative stress’s genotoxic role and determine the ameliorative effect of antioxidant therapy in LSD-affected cattle populations in Multan district, Pakistan. We performed correlation analysis and multivariable ordinal logistic regression to assess the precision of genotoxicity assessment and antioxidant therapy.

## 2. Materials and Methods

A 20-day prospective cohort study was conducted on sixty LSDV-affected cattle in Multan District, Pakistan. The individual animals were observed on day 0 to evaluate clinical and homeostatic changes (e.g., hematological, biochemical) due to the effects of LSD. The diseased cattle were divided into two categories: control and treated groups. As previously described, all the control and treated group animals were provided supportive treatment, including antipyretics, antihistaminics, and antibiotics [24]. To evaluate oxidative stress-induced genotoxicity in peripheral blood lymphocytes and the ameliorative effect of antioxidant therapy, the treated group was administered with Selevit Injection (Per ml containing Se 0.15 mg, Tocopherol 70 mg, Cyanocobalamin 0.1 mg, Adenosine 5-mono-phosphoric acid 5 mg, Fatro S.p.A, Ozzano dell’Emilia, Italy) at a dose rate of 10–20 mL/day for five consecutive days. The study was conducted in Multan District, Punjab, Pakistan (30.1575° N, 71.5249° E), and included various village cattle farms and dairy units. Initial indications of illness on the premises were documented approximately three months before the collection of samples, and physical assessments were carried out on affected livestock, as previously reported [25]. The ethical guidelines of The College of Veterinary Sciences, Riphah International University, Lahore, were strictly followed, and all the samples were collected after obtaining permission from the corresponding owners (Rcvets-1150).

The initial sampling from the animals was conducted on the day symptoms first appeared, with a final sampling performed after 20 days (Figure 1). We obtained blood specimens through jugular venipunctures. To avoid bias, a single veterinary professional executed all tests and analyses involved in this study.

For hematology analyses, the specimens were collected in vacutainers coated with EDTA. Meanwhile, the blood specimens without EDTA were centrifuged at 3000× *g* revolutions per minute (RPM) for 15 min and then stored at −20 °C for biochemical investigations [26]. Nodular lesions and scab samples were homogenized with a 50% phosphate-buffered saline (PBS) solution, diluted in 10% PBS, filtered through a 0.45 µM syringe filter, and stored at −80 °C until further use [11]. The viral DNA was extracted by GeneJET Viral DNA/RNA Purification Kit (Thermo Fisher Scientific, Waltham, MA, USA) per the manufacturer’s instructions. Previously reported primers targeting the viral attachment protein (RPO30 gene) (Forward 5′-TCTATGTCTTGATATGTGGTGGTAG-3′, and Reverse 5′-AGTGATTAGGTGGTGTATTATTTTCC-3′) were used [27]. The PCR products were visualized with an Omega FlourPlus Gel Documentation System (San Francisco, CA, USA). The positive samples were expected to produce a 172 bp amplicon.

An automatic hematology analyzer (Beckman Coulter, Brea, CA, USA) was used to evaluate the hematological indices. Some oxidative stress parameters, such as CAT, were determined by colorimetric method using a commercial kit (CAT100-1KT, Sigma-Aldrich^®^, St. Louis, MI, USA). Reduced (GSH) and NO were estimated according to [28,29,30], respectively. In the hemolysates, the SOD enzyme was assessed using the photooxidation method [31].

The malondialdehyde (MDA) levels in the plasma samples were quantified using a colorimetric reaction method of thiobarbituric acid (TBA) [32].

To perform the comet assay, lymphocytes were isolated from the sample. A mixture of 4 mL of Histopaque (lymphocyte separating media) and 3 mL of blood was prepared and centrifuged for 45 min at 800× *g* and 25 °C. The band of lymphocytes was then carefully aspirated and mixed with Roswell Park Memorial Institute 1640 Medium (RPMI). After a second round of centrifugation for 10 min at 250× *g*, the pallet produced was re-suspended in 1 mL PBS before the Neubauer chamber was used to count them [33]. Comet assay was conducted using the alkaline method [34]. A sample of 10 μL (0.1 × 10^5^ cells) was placed into 0.2 mL polypropylene tubes and mixed with 75 μL of low-melting-point agarose (LMPA in 0.7% PBS) at 37 °C [35], smeared on 76 mm × 26 mm microscope slides coated with 1.5% normal-melting-point agarose in PBS, and heated to 60 °C. The slides were then left to dry overnight at 22 °C. After the agarose had solidified for 10 min at 4 °C, the coverslips were removed, and the slides were immersed in a lysis solution (100 mM Na_2_EDTA, 2.5 M NaCl, 1% Triton X-100, 10% DMSO and 10 mM Tris-HCl) for 1 h at 4 °C. The slides were then set in a horizontal electrophoresis apparatus filled with a buffer comprising 300 mM NaOH and 1 mM Na_2_EDTA at 4 °C. The slides were incubated for 40 min in the buffer to unfold the DNA. The electrophoresis was performed at 25 Volts (0.83 V/cm) for 20 min and 300 mA in the dark to prevent DNA damage. The slides were washed thrice with neutralization buffer (0.4 M Tris-HCl, pH 7.5), followed by a single wash with 100% ethanol [36].

Slides were exposed to freezing water, prepared for microscopy, and then cells from the slides were examined. The comet tail length of the cells was used to categorize them into four groups: 0, intact cells; I, cells with a head diameter of the nucleus equal to or greater than the tail length; II, cells with a head diameter of the nucleus less than the tail length but with a tail length less than double head diameter; and III, cells with tail lengths more significant than twice their head diameter [37]. The slides were then analyzed using Casplab version 1.2.3 beta 2 for the comet head and tail parameters. The genetic damage index was calculated as described by Aroosa et al. [33].
$$\mathrm{GDI}=\frac{\left(No.\;of\;cells\;in\;class\;I\right)+\left(2\times No.\;of\;cells\;in\;class\;II\right)+\left(3\times No.\;of\;cells\;in\;class\;III\right)}{No.\;of\;cells\;in\;class\;0+No.\;of\;cells\;in\;class\;I+No.\;of\;cells\;in\;class\;II+No.\;of\;cells\;in\;class\;III}$$

The initial processing of the experimental findings was performed using Excel 2016 software. To compare the hematological and oxidative stress indices of control and treated animals at baseline and final sampling intervals through Minitab^®^ 21.3.1 (64-bit) software a two-way ANOVA assessed group differences (control, treated) and sampling time points (baseline and final). Pearson’s correlation test assessed the correlation among the various oxidative stress biomarkers. At the same time, the correlation between different oxidative stress biomarkers and GDI scores was analyzed through Spearman’s ranked correlation test. PCA was conducted to identify the pattern of the antioxidants and oxidative stress biomarkers dimensionality at the initial and final sampling of both groups. Indices with eigenvalues above 1.0 were classified as PC, with the higher one being known as the first PC [38]. An ordinal logistic regression model was employed to analyze the effect of oxidative stress parameters and antioxidant therapy on the likelihood of a change in genotoxicity using Minitab^®^ 21.3.1 (64-bit). A Pearson’s test was conducted to assess the goodness of fit of the model. The measure of association between the model’s response variable and the predicted variable was analyzed using Goodman–Kruskal Gamma. Furthermore, the Kruskal–Wallis test, followed by Dunn’s multiple comparison tests, was used to compare the mean rank sums of the comet assay scores among both groups at different sampling points using GraphPad Prism version 9.5. A *p*-value of less than 0.05 was considered to be statistically significant [39,40].

## 3. Results

### 3.1. Confirmation through PCR

The RPO30 gene PCR amplicon of 172 bp for confirmation of LSDV in different clinical samples was identified through agarose gel electrophoresis (Figure 2).

### 3.2. Hematological Findings

The results of the hematological findings for animals in both the control and treated groups at both initial and subsequent sampling intervals are presented in Table 1. 

At final sampling, the hematological parameters (RBCs, PCV and Hb) of treated individuals were significantly higher than that of the control animals (*p* < 0.05). The control group also improved the hematological indices at the final sampling time. However, the mean values of the control group regarding RBCs, Hb, and PCV were significantly less than the treated group at the end of the study. A significant increase (*p* < 0.05) in MCV was recorded in both groups at subsequent sampling compared to the initial values. The treated animals markedly improved their hematological indices compared to the control group. A notable difference in TLC levels between the control and treated animals was seen in the final samples. A statistically significant increase (*p* < 0.05) in lymphocyte count was observed in the control group compared to the treated animals in baseline values.

### 3.3. Evaluation of Oxidative Stress

The study’s results did not reveal any significant difference in the NO levels in the control animals at baseline and final sampling compared with those of the treated group at baseline sampling. Similarly, the mean MDA levels were recorded as significantly reduced in the treated group at final sampling compared to the levels at initial sampling in both groups and the final values of control animals. Conversely, considerably higher (*p* < 0.05) mean levels of GSH, CAT, and SOD were observed in the individuals of the treated group at final sampling. The results of the treated and control animals after both sampling periods are tabulated in Table 2.

### 3.4. Evaluation of Genotoxicity

The comet assay slides analysis revealed the highest percentage of class 2 and class 1 cells in control and treated groups samples at the baseline sampling. Figure 3 shows the comet assay of lymphocytes from various individuals of both groups.

At baseline sampling, an increased fraction of class 3 cells was observed in the control (7.54%) and treatment groups (6.66%) compared to the final interval. Table 3 describes the percentages (95% CI) of the cells of various classes observed during the comet assay. At the final sampling time, the control and treated groups revealed a significant (*p* < 0.05) decrease in GDI scores compared to their baseline values.

However, the antioxidant therapy showed a marked amelioration in the GDI in treated animals (Figure 4). The recorded medians score for GDI regarding the control group at baseline and final sampling were 0.79 (95%CI; 0.67, 0.9145) and 0.635 (95%CI; 0.5365, 0.777), respectively. The median GDI scores in the treated group at baseline and final sampling were 0.78 (95% CI; 0.701, 0.43) and 0.495 (95% CI; 0.8635, 0.6), respectively. Figure 3 shows the box and whiskers plot comparing mean rank scores of GDI in both groups at baseline and final sampling.

Correlation analysis of the various oxidative biomarkers and GDI in the infected individuals revealed a significant association among the variables. A significant correlation between serum NO (76%) and MDA levels (80%) was observed with GDI scores. On the other hand, antioxidant biomarkers, including GSH, CAT, and SOD, had a reasonably strong negative correlation (83%*,* 87%, and 84%, respectively) with GDI scores in animals of both control and treated groups (Table 4). 

In the results of PCA performed on the initial sampling data of the control group, PC1 exhibited the highest eigenvalue of 1.375, contributing approximately 27.49% of the total variance. PC2 followed closely with an eigenvalue of 1.194 (23.88% of the variance). For PC1, the NO had the highest positive contribution (0.3754), followed by GSH (0.4218), and then MDA, CAT, and SOD. Conversely, MDA had the highest positive contribution (0.3225) to PC2, with SOD (0.3664) also contributing significantly. The PCA of the treated group at initial sampling revealed the eigenvalues for PC1, PC2, PC3, PC4, and PC5 were 1.509, 1.138, 0.8747, 0.8564, and 0.6213, respectively. In PC1, the NO exhibited a minimal positive contribution (0.0011), whereas MDA and GSH showed notable contributions of 0.2565 and 0.1308, respectively. CAT contributed significantly (0.3775), and SOD also had a discernible positive influence (0.2341). PC2 was primarily influenced by the NO (0.6522). The eigenvalues for each PC in the control group at the final sampling stage were as follows; PC1 (1.470), PC2 (1.274), PC3 (1.200), PC4 (0.7659), and PC5 (0.2904). For PC1, the NO had a substantial contribution (0.3557), followed by GSH (0.1370) and SOD (0.4496). Meanwhile, in PC2, GSH exhibited the highest contribution (0.4471), followed by NO (0.3989) and MDA (0.1511). The PCA summary of the treated group at final sampling data revealed that PC1 accounted for 28.59% of the total variance (eigenvalue of 1.429). PC2 represented 23.78% of the variance, while PC3, PC4, and PC5 accounted for 17.70%, 16.25%, and 13.69%, respectively. In PC1, SOD (0.3632) was recorded as the most significant variable, followed by CAT (0.2906) and GSH (0.0816). Conversely, PC2 was predominantly influenced by CAT (0.4362), MDA (0.2056), and NO (0.0693) (Figure 5).

The multivariable ordinal logistic regression results regarding the influence of NO, MDA, GSH, CAT, and SOD on GDI did not show a significant (*p* > 0.05) effect in the model. However, the bivariable model analysis revealed that each unit increase in NO and MDA resulted in a 13% (OR; 0.87 95%CI; 77–99 and 95%CI; 82–93, respectively) increase in the genotoxicity of infected individuals. Based on the results of the Pearson test (*p =* 1.000), the model analysis regarding NO and MDA revealed a statistically significant impact. Moreover, the ordinal logistic model regarding the use of antioxidant therapy revealed 73.95-times (95%CI; 17.36–314.96) increase in the chances of amelioration in the GDI score in treated animals as compared to the control animals. Pearson’s test showed the model’s goodness of fit (*p =* 0.843), whereas the Goodman–Kruskal gamma value (0.91) defined the perfect association between the response variable and the predicted probabilities. 

## 4. Discussion

LSD has been classified as a transboundary livestock disease because it has the potential to spread between countries and have significant economic, trade, and food security implications [41]. The present study focuses on the correlation of oxidative stress markers with the extent of DNA damage using comet assay parameters and the ameliorative use of Se and vitamin E antioxidants in LSD-affected cattle under natural conditions in Pakistan. 

PCR is the most reliable technique for confirming the presence of LSDV and has been found to have a high sensitivity when detecting its genome in skin nodules [7]. The high level of virus detection through PCR testing may be explained by the virus’ tendency to affect the skin and its ability to remain at high concentrations in cutaneous tissues [13]. In addition, the infected animals had depleted leukocytes, lymphocytes, monocytes, and eosinophils, which suggests a viral infection [42]. Excessive corticosteroid production decreases the number of lymphocytes [41]. The present study’s findings align with previous results [6,13]. Previous studies have also observed thrombocytopenia mainly due to shortened platelet lifespan [13,25]. Systemic vasculitis is often the underlying trigger in LSDV infection due to endothelial damage leading to excessive platelet consumption. 

In response to viral infection, the activation of cellular stress mechanisms in infected cells is a complicated process that simultaneously stimulates cell survival and death mechanisms. Poxviruses utilize the cellular de-novo biosynthesis of fatty acids, with a particular emphasis on the synthesis of palmitates. Each of these molecules undergo the process of beta-oxidation within the mitochondria. Additionally, in conjunction with the catabolism of glutamine, they produce acetyl-CoA and alpha-ketoglutarate, respectively [43]. Both molecules are utilized as substantial energy sources within the infected cells as substitutes for the glucose TCA cycle [44,45,46]. Through this metabolic pathway, oxygen plays a crucial role as the primary electron acceptor in the electron transport chain of oxidative phosphorylation. This process generates water (H_2_O) and other ROS [47,48]. ROS induce detrimental effects on cellular structures, such as protein denaturation, lipoperoxidation, and DNA degradation. However, ROS can serve as a secondary messenger in the modulation of inflammation, promoting cell proliferation and regulating apoptosis, all of which contribute to maintaining the cellular homeostasis [49,50,51,52]. Due to cytotoxic activity, antioxidant enzymes and vitamins tightly limit cellular ROS levels. These mechanisms are called oxidative stress responses [48].

Meanwhile, elevated nitric oxide (NO) production is typically attributed to the induction of inducible NO synthase (iNOS), which is frequently expressed by immunological phagocytes, epithelial and neuronal cells [53,54,55]. iNOS has the potential to release an increased and sustained amount of nitric oxide compared to neuronal NOS or endothelial NOS. iNOS is expressed ubiquitously during in vivo viral replication. The induction of iNOS during virus infection is upregulated by pro-inflammatory cytokines such as interferon-γ (IFN-γ) [50]. Furthermore, IFN-γ is linked to activating type 1 helper T cell (Th1) responses [56]. The outcomes show that IFN-γ is essential in inducing the overexpression of iNOS and NO, which are involved in the progression of viral infection [57]. MDA is the primary lipid peroxidation marker extensively explored as an oxidative stress biomarker. The extent of oxidative stress can be determined by measuring the concentration of MDA [58,59]. Endogenous aldehydes, such as MDA, are highly reactive and capable of forming aldehyde-derived DNA adducts, which result in DNA damage [60]. 

So far no research study has worked on the relationship between NO and MDA levels with DNA damage as reflected in the comet assay. Our results showed a positive relationship between serum levels of NO and MDA with the GDI, thus suggesting that LSDV-induced oxidative stress may result in DNA damage in infected animals. The GDI scores of the individuals revealed a strong correlation (80%) with serum MDA and (76%) NO levels, indicating that oxidative stress-induced MDA production in LSD is directly linked with genotoxicity in the infected animals. Moreover, the results are supported by the ordinal logistic regression model finding, which showed that each unit increase in the serum level of NO and MDA results in a 13% increase in the odds of genotoxicity in infected animals, respectively. Genotoxicity was also negatively correlated with the antioxidant capacity of the cattle. Serum GSH, SOD, and CAT levels were negatively associated (83%, 84%, and 87%, respectively) with genetic damage. 

Vitamin E’s primary function as an antioxidant is to capture lipid peroxyl radicals, which initiate and propagate lipid oxidation [61,62]. Se is a structural component of antioxidant enzymes, such as glutathione peroxidase and thioredoxin. It counters the toxic effects of some toxic chemicals and protects the tissues against free radical injury [63,64]. Furthermore, in the present study, the combined therapy of animals with Se and vitamin E was effective against oxidative stress and genotoxicity in the LSD-affected animals. Results of ordinal logistic regression revealed that using antioxidant therapy results in almost 74-times increased odds of amelioration in the genotoxicity compared to the individuals of the control group. The results are consistent with previous studies, which showed that Vitamin E and Se can significantly reduce oxidative stress-induced genetic damage to peripheral lymphocytes [65,66,67,68]. 

## 5. Conclusions

In conclusion, our study revealed that LSD-induced oxidative stress and lipid peroxidation products including NO and MDA, cause genotoxicity in the peripheral blood lymphocytes of affected animals. It is further suggested that the intervention of antioxidant therapy combined with supportive treatment results in the significant amelioration of the hematological indices, oxidative stress, and genotoxicity in LSD-affected cattle.

## Figures and Tables

**Figure 1 vetsci-10-00643-f001:**
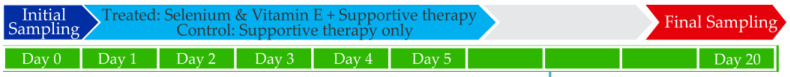
Indicates the timeline of the experimental procedure including initial sampling at day 0 followed by the treatment of infected animals for five consective days and final sampling at day 20.

**Figure 2 vetsci-10-00643-f002:**
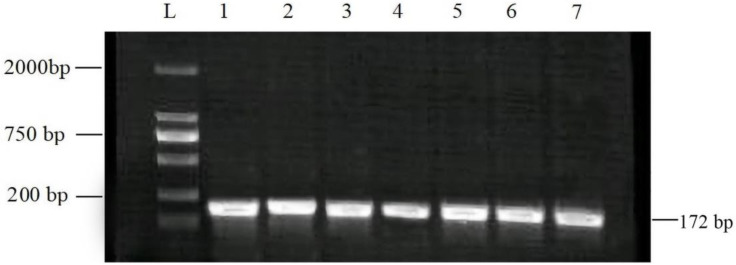
Agarose gel electrophoresis pattern of PCR assay using the RPO30 gene to detect LSDV. Lanes 1–6 of the gel show 172 bp PCR products from field samples, while lane 7 is a positive control. L: molecular ladder marker was used as a reference.

**Figure 3 vetsci-10-00643-f003:**
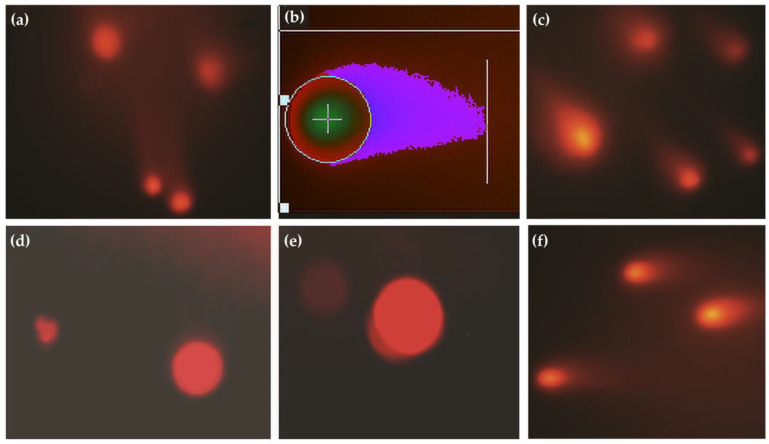
(**a**) Comet images of lymphocytes in the control group, (**b**) analysis of comet in diseased animals using Casplab software, (**c**) individuals of the treated group at initial sampling, (**d**,**e**) lymphocytes of animals in the treated group after therapy, (**f**) comet micrograph of lymphocytes from an individual of the control group at final sampling.

**Figure 4 vetsci-10-00643-f004:**
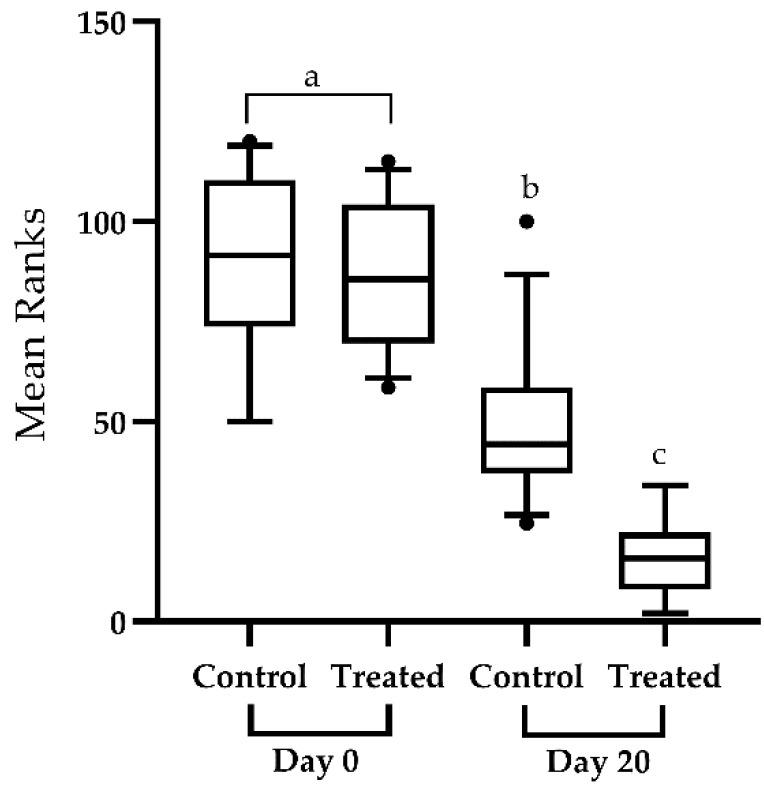
Box and whiskers plot showing 5–95 percentile (box shows the first and third quartiles) rank-sums for the genetic damage index in the control (*n* = 30) and treated groups (*n* = 30). The genetic damage index was assessed. The comparison of median scores was performed through Kruskal–-Wallis test and Dunn’s multiple comparison test. Plots with various superscripts differ substantially from one another (*p* < 0.05). Significant decrease in the genetic damage index scores of treated individuals at day-20 of the study compared to the control group and baseline sampling.

**Figure 5 vetsci-10-00643-f005:**
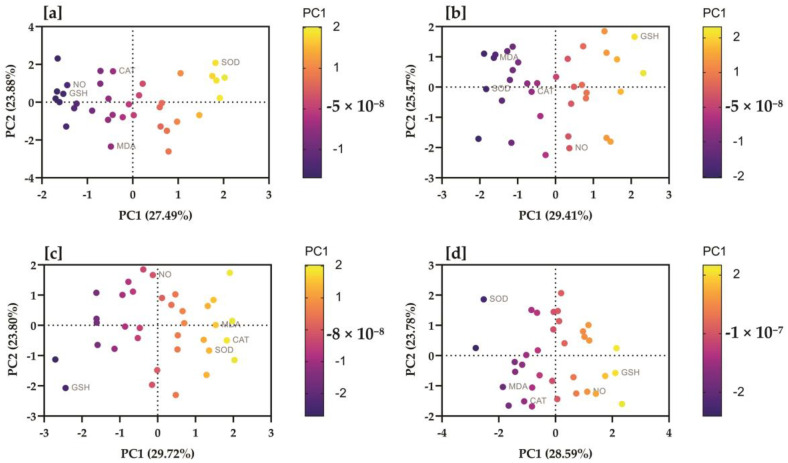
Serum nitric oxide, malondialdehyde, glutatione, catalase, and superoxide dismutasewere correlated with study groups at different time intervals. The proportion of variances of each PC (based on eigenvalues) is presented in parentheses, along with the X and Y coordinates. Biplots of principal component analysis showing PC1 and PC2 of the data regarding (**a**) the control group analyzed at initial sampling, (**b**) the control group at final sampling, (**c**) the treatment group at initial sampling, (**d**) the treatment group at final sampling. Colored by the strength of the relationship of the variable with the PC (strong positive = yellow; strong negative = dark blue).

**Table 1 vetsci-10-00643-t001:** Means without common superscript letters of hematological parameter values between the control and treated groups at baseline and final sampling points were compared using two-way ANOVA followed by Tukey’s post hoc test (*p* < 0.05).

Variable	Control (*n* = 30)		Treated (*n* = 30)	
	Initial Sampling	Final Sampling	Initial Sampling	Final Sampling
RBCs × 10^6^/µL	5.37 ± 0.021 ^c^	5.57 ± 0.012 ^b^	5.41 ± 0.019 ^c^	5.71 ± 0.025 ^a^
Hb gm/dL	8.88 ± 0.022 ^b^	9.11 ± 0.01 ^a^	8.89 ± 0.022 ^b^	9.17 ± 0.019 ^a^
PCV %	28.7 ± 0.026 ^b^	31.2 ± 0.023 ^a^	28.6 ± 0.027 ^b^	31.2 ± 0.02 ^a^
MCV fl	53.4 ± 0.222 ^c^	56 ± 0.127 ^a^	52.9 ± 0.193 ^c^	54.8 ± 0.234 ^b^
MCHC %	31 ± 0.090 ^a^	29.2 ± 0.039 ^b^	31.1 ± 0.081 ^a^	29.4 ± 0.060 ^b^
Platelets × 10^3^/µL	132 ± 0.165 ^b^	112 ± 0.139 ^c^	132 ± 0.165 ^b^	141 ± 0.172 ^a^
TLC × 10^3^/µL	8.74 ± 0.027 ^b^	9.71 ± 0.011 ^a^	8.77 ± 0.026 ^b^	6.25 ± 0.024 ^c^
Neutrophil × 10^3^/µL	3.53 ± 0.011 ^a^	3.45 ± 0.004 ^b^	3.54 ± 0.01 ^a^	2.22 ± 0.009 ^c^
Eosinophil × 10^3^/µL	0.138 ± 0.0004 ^b^	0.153 ± 0.0001 ^a^	0.139 ± 0.0004 ^b^	0.0988 ± 0.0003 ^c^
Lymphocyte × 10^3^/µL	4.81 ± 0.015 ^b^	5.84 ± 0.006 ^a^	4.82 ± 0.014 ^b^	3.78 ± 0.014 ^c^
Monocyte × 10^3^/µL	0.267 ± 0.0008 ^a^	0.259 ± 0.0003 ^b^	0.268 ± 0.0007 ^a^	0.156 ± 0.0006 ^c^

**Table 2 vetsci-10-00643-t002:** Comparison of the values of oxidative stress biomarkers in control and treated groups at baseline and final sampling. The mean values (±SEM) not sharing a common superscript letter are significantly different (*p* < 0.05).

Variable	Control (*n* = 30)		Treated (*n* = 30)	
	Initial Sampling	Final Sampling	Initial Sampling	Final Sampling
NO (ng/μL)	52.8 ± 0.161 ^a^	53 ± 0.15 ^a^	52.5 ± 0.165 ^a^	38.5 ± 0.147 ^b^
MDA (nmol/mL)	62.2 ± 0.332 ^a^	50.6 ± 0.203 ^b^	50.6 ± 0.225 ^b^	32.3 ± 0.218 ^c^
GSH (mg/dL)	5.47 ± 0.048 ^c^	6.43 ± 0.050 ^b^	5.56 ± 0.047 ^c^	11.3 ± 0.14 ^a^
CAT (U/L)	211 ± 0.161 ^c^	241 ± 0.163 ^b^	211 ± 0.177 ^c^	313 ± 0.257 ^a^
SOD (U/mL)	7.41 ± 0.049 ^c^	8.45 ± 0.05 ^b^	7.51 ± 0.044 ^c^	13.5 ± 0.047 ^a^

**Table 3 vetsci-10-00643-t003:** Percentages (95% confidence interval) of the cells counted as oxidative stress biomarkers in control and treated groups at baseline and final sampling.

Groups	Sampling Interval	Row-Wise Percentages (95% Confidence Interval) of Cells Counted for Genetic Damage			
		Class 0	Class 1	Class 2	Class 3
Control (*n* = 30)	Initial	21.10%	35.28%	36.10%	7.54%
		(19.1–23.2)	(32.8–37.8)	(33.7–38.6)	(6.2–9)
	Final	34.82%	35.71%	22.71%	6.78%
		(32.4–37.3)	(33.3–38.2)	(20.6–24.9)	(5.6–8.2)
Treated (*n* = 30)	Initial	22.02%	35.33%	36.01%	6.66%
		(19.9–24.2)	(32.9–37.8)	(33.6–38.5)	(5.5–8)
	Final	50.28%	32.54%	17.19%	0%
		(47.7–52.8)	(30.1–34.9)	(15.3–19.2)	(0–0.2)

**Table 4 vetsci-10-00643-t004:** Matrix depicts the correlation between all the possible pairs of values of oxidative stress biomarkers and genetic damage in the blood lymphocytes of infected individuals. Pearson’s correlation test correlated the continuous variables. In contrast, Spearman’s ranked correlation was used to correlate the variables with the genetic damage index. *p* < 0.05 was considered significant. R^2^ values of 0.90 to 1.00, 0.70 to 0.90, 0.50 to 0.70, 0.30 to 0.50, and 0.00 to 0.30 represent very high correlation, high correlation, moderate correlation, low correlation, and negligible correlation, respectively. A negative sign (-) with an R^2^ value describes the negative correlation between the two variables.

Parameters	NO (ng/μL)	MDA (nmol/mL)	GSH (mg/dL)	CAT (U/L)	SOD (U/mL)	GDI
NO (ng/μL)	-	0.89	−0.97	−0.95	−0.97	0.76
MDA (nmol/mL)	0.89	-	−0.91	−0.92	−0.92	0.8
GSH (mg/dL)	−0.97	−0.91	-	0.98	0.98	−0.83
CAT (U/L)	−0.95	−0.92	0.98	-	0.99	−0.87
SOD (U/mL)	−0.97	−0.92	0.98	0.99	-	−0.84
GDI	0.76	0.8	−0.83	−0.87	−0.84	-

## Data Availability

The data presented in this study are available in the Appendix A.

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
