# Peer review of "Unveiling Oxidative Stress-Induced Genotoxicity and Its Alleviation through Selenium and Vitamin E Therapy in Naturally Infected Cattle with Lumpy Skin Disease"

_vetsci, 2023, doi:10.3390/vetsci10110643_

Round 1

Reviewer 1 Report

Comments and Suggestions for Authors

The study has exploring the lumpy skin disease causes-oxidative stress-induced genotoxicity. Furthermore, the study also has assess the its alleviation through selenium and vitamin E therapy in naturally infected cattle. The topic is interesting. The experimental design is well. The following revision would improve the quality of the paper. 

1) Please state the objective of the study in the Abstract. 

2) Please write the P value in italic throughout the paper.

3) Line 32, 'SOD, CAT, etc.'please write all the abbreviation with their full name when it was firstly appeared in the manuscript.

4)  Lines 72-73, it is too simple to description the background of selenium and VitE in the introduction. Please add some new refernece about them. Such as: Selenium-Enriched Cardamine violifolia Increases Selenium and Decreases Cholesterol Concentrations in Liver and Pectoral Muscle of Broilers. The Journal of Nutrition3. 2022. Role of Selenium and Vitamins E and B9 in the Alleviation of Bovine Mastitis during the Periparturient Period. Antioxidants 2022. On the Potential Role of the Antioxidant Couple Vitamin E/Selenium Taken by the Oral Route in Skin and Hair Health. Antioxidants 2022. 

5) Please simplify the photo in Figure 1 by cutting out unnecessary parts. 

6) Please improve the presentation quality of the Table 1 and others. Such as too many unnecessary decimals.

7)Please add the replicates n=? and statistical significant setting in the footnotes or figure legends throughout the paper.

8) Please add the full name for the abbrevations in the footnotes or figure legends. 

9) Please check the references and make sure they followed the journal style.

Author Response

The study has exploring the lumpy skin disease causes-oxidative stress-induced genotoxicity. Furthermore, the study also has assess the its alleviation through selenium and vitamin E therapy in naturally infected cattle. The topic is interesting. The experimental design is well. The following revision would improve the quality of the paper. 

Response: Thank you very much for taking the time to review this manuscript. Please find the detailed responses below and the corresponding revisions/corrections highlighted/in track changes in the re-submitted files.

1) Please state the objective of the study in the Abstract. 

Response: Thanks for the comment. We have updated the objective of the study in abstract section stating that “A 20-day prospective cohort study was done on sixty infected cattle with the aim to evaluate LSD-induced oxidative stress's genotoxic role and to determine the ameliorative effect of antioxidant therapy using principal component analysis (PCA) and a multivariable ordinal logistic regression model”.

2) Please write the P value in italic throughout the paper.

Response: Thanks for the comment. We have modified the formatting according to the guideline.

3) Line 32, 'SOD, CAT, etc.'please write all the abbreviation with their full name when it was firstly appeared in the manuscript.

Response: Thanks for highlighting the shortcoming. We have corrected the manuscript with the full name given with the abbreviation where first it appears in the text.

4)  Lines 72-73, it is too simple to description the background of selenium and VitE in the introduction. Please add some new refernece about them. Such as: Selenium-Enriched Cardamine violifolia Increases Selenium and Decreases Cholesterol Concentrations in Liver and Pectoral Muscle of Broilers. The Journal of Nutrition3. 2022. Role of Selenium and Vitamins E and B9 in the Alleviation of Bovine Mastitis during the Periparturient Period. Antioxidants 2022. On the Potential Role of the Antioxidant Couple Vitamin E/Selenium Taken by the Oral Route in Skin and Hair Health. Antioxidants 2022. 

Response: Thanks for the guidance. We have updated the introduction section with the new references as suggested by the respected reviewer. The references have been added in the paragraph “Among the various antioxidants, vitamin E and selenium (Se) exhibit a unique feature that they function synergistically to combat lipid peroxides (ROOH). Vitamin E can directly engage with lipid free radicals (ROO•). While, Se is an essential trace mineral [19] which serves as the principal co-factor for glutathione peroxidase (GPx) which is an essential enzyme facilitating the conversion of lipid peroxides into inert compounds [20]. Using these natural antioxidants, including vitamin E and Se, has been shown to mitigate the negative consequences of the overproduction of ROS [21]. Both Se and vitamin E have been specifically aimed at enhancing the metabolism, reducing oxidative stress, and bolstering the immune and anti-inflammatory responses in dairy cattle [22]. Vitamin E has been also been reported with the capability of reducing genotoxic effects [23]”.

5) Please simplify the photo in Figure 1 by cutting out unnecessary parts. 

Response: The authors have replaced the figure 1 with unnecessary parts cropped and corrected picture of gel electrophoresis.

6) Please improve the presentation quality of the Table 1 and others. Such as too many unnecessary decimals.

Response: Authors are thankful to the reviewer for pointing out this shortcoming. We have rounded off all the decimal in the table throughout the manuscript.  

7)Please add the replicates n=? and statistically significant setting in the footnotes or figure legends throughout the paper.

Response: Replicates number (n=) have been added along with statistically significant setting in the figure legends throughout the paper.

8) Please add the full name for the abbrevations in the footnotes or figure legends. 

Response: Abbreviations have been updated with the full name in the figure legends.

9) Please check the references and make sure they followed the journal style.

Response: Thank you for the feedback. We have checked and verified the reference style according to the journal’s guidelines.

Reviewer 2 Report

Comments and Suggestions for Authors

The paper is well written, clearly useful. Thanks for the contribute

Author Response

The paper is well written, clearly useful. Thanks for the contribute.

Response: Thank you very much for sparing your precious time to review this manuscript. We are highly obliged to have such words from the reviewer. 

Reviewer 3 Report

Comments and Suggestions for Authors

The authors in this manuscript purpose to research the effect of antioxidants´therapy on LSD disease.

For clarification: please indicate in a timeline the experimental parameters because in the M&M there are additional time periods indicated, but there´s no further mention in the text of the manuscript.

Additionally, it would be interesting to know how the therapy improved the health parameters of the cattle with LSD after 20 days. In principle, it would be interesting to know if the therapy has impacts on the prognostic health outcome. 

Do you have any genetic analysis of the virus type that was present in the cattle?

Minor corrections:

Please, list all abbreviations with full name in a Table. And indicate full name with first occurrence of abbreviation.

M&M must be better structured.

Comments on the Quality of English Language

-

Author Response

The authors in this manuscript purpose to research the effect of antioxidants´therapy on LSD disease.

For clarification: please indicate in a timeline the experimental parameters because in the M&M there are additional time periods indicated, but there´s no further mention in the text of the manuscript.

Additionally, it would be interesting to know how the therapy improved the health parameters of the cattle with LSD after 20 days. In principle, it would be interesting to know if the therapy has impacts on the prognostic health outcome. 

Response: Thank you for your valuable feedback. We have added the timeline of the experimental procedure in the methodology section. Regarding your comment on the impact of therapy on the prognostic health outcome, we would like to clarify that our study was not designed to assess long-term prognostic outcomes using an experimental model. However, we did observe significant improvements in the health status and recovery rate of the treated animals compared to the control group over a 20-day period. While we acknowledge the importance of evaluating long-term prognostic outcomes, our study primarily aimed to investigate the immediate effects of the therapy within a 20-day timeframe.

Do you have any genetic analysis of the virus type that was present in the cattle?

 Response: Thanks for the comment. We have sent the PCR product of RPO30 gene for sequencing, hopefully, we will receive the nucleotide sequencing file within a couple of weeks. Moreover, we are also working on another manuscript regarding the genetic analysis and molecular characterization of the Lumpy skin disease virus isolated from various regions of the Punjab Province of Pakistan.

Minor corrections:

Please, list all abbreviations with full name in a Table. And indicate full name with first occurrence of abbreviation.

 Response: we have provided a table stating the list of abbreviation along with the full form in the revised manuscript.

M&M must be better structured.

Response: The authors respect the comment of the reviewer. The authors have already structured the methodology section in a systematic way. However, as the respected reviewer has highlighted a timeline of the experimental process is missing in the methodology section. The authors are highly thankful to the reviewer for describing the shortcomings. We have added the timeline of experimental procedure in the methodology section.

Round 2

Reviewer 3 Report

Comments and Suggestions for Authors

You need to indicate what lane 6 in Figure 2 signifies before publication.

Author Response

Thank you very much for the valuable comment. We apologize for overseeing typo error. Along with the Lane 1-5, Lane 6 also represented the PCR amplicon of LSD infected animals. We have corrected the legend of the figure 2 as “Figure 2. Agarose gel electrophoresis pattern of PCR assay using the RPO30 gene to detect LSDV. Lanes 1-6 of the gel show 172bp PCR products from field samples, while lane 7 is a positive control. L: molecular ladder marker was used as a reference”